# Genetic markers of abdominal obesity and weight loss after gastric bypass surgery

**Martin Aasbrenn**[1,2☯*], **Mathilde Svendstrup**[1,3,4☯¤], **Theresia M. Schnurr**[1], **Dorte Lindqvist Hansen**[4], **Dorte Worm**[5], **Marie Balslev-Harder**[1], **Niels Grarup**[1], **Kristoffer Sølvsten Burgdorf**[6], **Henrik Vestergaard**[1,7], **Oluf Pedersen**[1], **Lars Ängquist**[1], **Mogens Fenger**[8], **Thorkild I. A. Sørensen**[1,9], **Sten Madsbad**[10], **Torben Hansen**[1]

1 Novo Nordisk Foundation Center for Basic Metabolic Research, Faculty of Health and Medical Sciences, University of Copenhagen, Copenhagen, Denmark, 2 Geriatric Research Unit, Department of Geriatrics, Bispebjerg-Frederiksberg University Hospital, Copenhagen, Denmark, 3 Department of Endocrinology, Bispebjerg-Frederiksberg University Hospital, Copenhagen, Denmark, 4 Steno Diabetes Center Copenhagen, Gentofte, Denmark, 5 Department of Medicine, Amager Hospital, Copenhagen, Denmark, 6 Department of Clinical Immunology, Copenhagen University Hospital, Copenhagen, Denmark, 7 Bornholms Hospital, Rønne, Denmark, 8 Department of Clinical Biochemistry, Hvidovre University Hospital, Copenhagen, Denmark, 9 Department of Public Health, University of Copenhagen, Copenhagen, Denmark, 10 Department of Endocrinology, Hvidovre University Hospital, Copenhagen, Denmark

☯ These authors contributed equally to this work.
¤ Current address: Department of Endocrinology, Copenhagen University Hospital, Copenhagen, Denmark
* Martin.Aasbrenn@gmail.com

**Data Availability Statement:** The data set contains potentially identifying and sensitive genetic information on a vulnerable population, and the permission from the Scientific Ethics Committee of

## Abstract

### Background

Weight loss after bariatric surgery varies widely between individuals, partly due to genetic differences. In addition, genetic determinants of abdominal obesity have been shown to attenuate weight loss after dietary intervention with special attention paid to the rs1358980-T risk allele in the *VEGFA* locus. Here we aimed to test if updated genetic risk scores (GRSs) for adiposity measures and the rs1358980-T risk allele are linked with weight loss following gastric bypass surgery.

### Methods

Five hundred seventy six patients with morbid obesity underwent Roux-en-Y gastric bypass. A GRS for BMI and a GRS for waist-hip-ratio adjusted for BMI (proxy for abdominal obesity), respectively, were constructed. All patients were genotyped for the rs1358980-T risk allele. Associations between the genetic determinants and weight loss after bariatric surgery were evaluated.

### Results

The GRS for BMI was not associated with weight loss (β = -2.0 kg/100 risk alleles, 95% CI -7.5 to 3.3, p = 0.45). Even though the GRS for abdominal obesity was associated with an attenuated weight loss response adjusted for age, sex and center (β = -14.6 kg/100 risk alleles, 95% CI -25.4 to -3.8, p = 0.008), it was not significantly associated with weight loss after adjustment for baseline BMI (β = -7.9 kg/100 risk alleles, 95% CI -17.5 to 1.6, p = 0.11).

the Capital Region, Denmark from 2009 did not include acceptance of sharing of data. Anonymized patient level data can only legally be accessed upon acceptance from The Danish Data Protection Agency (www.datatilsynet.dk; dt@datatilsynet.dk), the goverment body that regulates the use of Danish research data. Interested parties can contact the corresponding author (MA) for further information.

**Funding:** This study was funded by a grant from The Ministry of Higher Education and Science The UNIK Initiative: Food, Fitness & Pharma for Health and Disease (TH, http://ufm.dk). The project was also supported with funds from the Challenge Programme MicrobLiver (Grant No. NNF15OC0016692) (TH, http://challenge.novonordiskfoundation.com). The Novo Nordisk Foundation Center for Basic Metabolic Research is an independent research centre at the University of Copenhagen and is partly funded by an unrestricted donation from the Novo Nordisk Foundation.

**Competing interests:** The authors have declared that no competing interests exist.

Similarly, the rs1358980-T risk allele was not significantly associated with weight loss ($\beta$ = -0.8 kg/risk allele, 95% CI -2.2 to 0.6, p = 0.25).

## Discussion

GRSs for adiposity derived from large meta-analyses and the rs1358980-T risk allele in the *VEGFA* locus did not predict weight loss after gastric bypass surgery. The association between a GRS for abdominal obesity and the response to bariatric surgery may be dependent on the association between the GRS and baseline BMI.

## Introduction

Bariatric surgery is the most effective treatment for morbid obesity [1]. However, the weight loss response after bariatric surgery varies widely between individuals [2]. A part of this variation is probably due to genetic factors, as close biological relatives tend to have quite similar responses to weight loss interventions [3–5]. Potential specific genetic determinants of the individual variation in weight loss after bariatric surgery have been explored in some studies [5–13] with limited success.

Whether known adiposity-related risk single nucleotide polymorphisms (SNPs) are related to the response to weight loss interventions has been studied in recent years [10, 11, 14]. Genetic risk scores (GRSs) related to body mass index (BMI) based on 33 and 77 SNPs were not associated with the response to bariatric surgery in studies undertaken from 2014 and 2019 [11, 14]. In meta-analysis of GWASs from 2018 for BMI outcome in population studies, a total of 941 SNPs were identified [15]. Whether a GRS constructed from these 941 SNPs associates with the response to weight loss after bariatric surgery has not been investigated.

We previously replicated the association of a GRS based on 3 SNPs for abdominal obesity (waist-hip-ratio adjusted for BMI; WHR$_{BMI}$) with increased weight loss response after bariatric surgery [6, 11]. Conversely, in another study including an updated GRS of 49 SNPs associated with WHR$_{BMI}$, the genetic determinants were linked with attenuated weight loss after dietary intervention in women with special attention paid to the rs1358980-T risk allele [16]. The rs1358980-T risk allele is located upstream to a well-known angiogenesis gene; *VEGFA*, that codes for the vascular endothelial growth factor A (VEGF-A). The total number of SNPs known to be related to abdominal obesity has also increased rapidly. Thus in a recent meta-analysis, 346 loci were identified for the specific phenotype WHR$_{BMI}$ [17]. An updated GRS based on these 346 SNPs has not been investigated in relation to the response to weight loss after bariatric surgery.

The primary aim of this study was to examine the effect of recently developed GRSs for BMI and WHR$_{BMI}$, respectively, on the inter-individual variations in weight loss after bariatric surgery. The secondary aim was to similarly examine the association of rs1358980-T risk allele in the *VEGFA* locus with weight loss after bariatric surgery.

## Materials and methods

### Study design and setting

Danish adult patients $\geq$ 25 years of age with morbid obesity (BMI > 50 kg/m$^2$ or BMI > 35 kg/m$^2$ with obesity-related complications) being offered bariatric surgery were included. Possible obesity-related complications included hypertension, diabetes mellitus, sleep apnea,

respiratory failure or osteoarthrosis of weight-bearing joints. All the included patients were operated with Roux-en-Y gastric bypass surgery, created with a 25 ml gastric pouch, a 75 cm long bilio-pancreatic limb and a Roux limb of 100 cm [18].

From June 2009 to April 2013, patients with morbid obesity previously referred to a public obesity center at Hvidovre Hospital, Capital Region of Denmark, were invited to participate. Information was collected at visits to the hospital before, 1 year after (mean 392 days after the surgery, standard deviation (SD) 59 days) and 2 years after Roux-en-Y gastric bypass surgery (mean 776 days after the surgery, SD 71 days). Between December 2007 and November 2009, adult patients referred to the Hamlet Hospital, Søborg, Capital Region of Denmark, were invited to participate. The patients operated at Hamlet Hospital were evaluated at Steno Diabetes hospital 1 year after surgery (mean 371 days after the surgery, SD 26 days) and at a second control 2–4 years after Roux-en-Y gastric bypass surgery (mean 838 days after the surgery, SD 319 days, interquartile range 531 days). In total, 1511 patients agreed to participate. All study visits included blood sampling and measurement of body weight and height.

## Variables

**Genotyping.** DNA was extracted from blood leukocytes at LGC Genomics (LGC, Middlesex, United Kingdom), and 1511 samples from all the participants at inclusion (n = 1414) were genotyped by the *Illumina HumanCoreExome Beadchip* (Illumina, San Diego, CA) using Illumina's *HiScan* system at the Novo Nordisk Foundation Center for Basic Metabolic Research in Copenhagen, Denmark. Genotypes were called using the Genotyping module (version 1.9.4) of *GenomeStudio* software (version 2011.1, Illumina). During quality control, we excluded samples that were first degree relatives (n = 34), ethnic outliers (n = 72), had extreme inbreeding coefficients (n = 99), mislabeled gender (n = 8) or a call rate < 95% (n = 177), leaving 1024 individuals. Among these genotyped individuals, patients who did not undergo bariatric surgery or did not have the required follow-up visits were excluded, leaving 576 individuals in the main analysis; 351 recruited at Hvidovre Hospital and 225 recruited at Hamlet Hospital. SNPs that were used for the calculation of the GRSs were either directly genotyped or imputed to the Haplotype Reference Consortium panel (HRC, version 1.1). The imputation quality was high (INFO > 0.95) for all imputed variants included in the GRS constructions. No included variants were significantly deviating from Hardy-Weinberg equilibrium (p > 0.05).

**GRS construction.** We constructed weighted GRSs by summing the number of risk-increasing alleles weighted by the effect size of the variant estimated in the GWAS discovery studies [15, 17]. We then normalized the weighted GRS to make effect sizes of the weighted GRSs comparable to risk allele effects [19]. The $GRS\text{-}BMI_{941}$ for BMI was based on 941 SNPs associated with cross-sectional BMI in European populations [15]. The $GRS\text{-}WHR_{344}$ was based on 344 (out of 346) SNPs that were associated with $WHR_{BMI}$ in the discovery meta-analysis [17]. We only included 344 (out of the 346 SNPs) because two SNPs (rs805768 and rs7798002) were non-biallelic and therefore excluded from the GRS calculation.

## Anthropometrics

Body weight was measured at the first visit to the obesity center (initial body weight) and at the clinical controls after bariatric surgery [19]. The lowest body weight at a post-surgical clinical visit was used as nadir body weight in the analyses. BMI (kg/m²) was calculated as the body weight (kg) divided by the height (m) squared [20]. WHR was measured at the first visit to the obesity center in a subset of 272 patients. Waist circumference was measured in cm at the anatomical location with the largest circumference, hip circumference was measured at the largest circumference around the buttocks and waist-hip-ratio (WHR) was calculated as [waist

circumference] / [hip circumference]. Weight loss was calculated as [initial weight]–[nadir weight].

## Statistical analysis

Baseline data are presented as SDs or proportions (percentages).

The associations between the three genetic determinants (GRS-BMI$_{941}$, GRS-WHR$_{344}$, *VEGFA*-rs1358980-T) and baseline BMI and WHR, respectively, were examined with linear regression (WHR was adjusted for baseline BMI to derive WHR$_{BMI}$). The associations between weight loss after bariatric surgery and BMI and WHR$_{BMI}$ were also examined with linear regression.

To answer the primary aim, the GRS-BMI$_{941}$ and the GRS-WHR$_{344}$ were used as explanatory variables in multiple linear regression analyses with weight loss as dependent variable. To answer the secondary aim, *VEGFA*-rs1358980-T was used in the same fashion. These analyses are presented adjusted for baseline age, sex and recruitment center and in subsequent analyses additionally adjusted for baseline BMI. Analyses are also presented stratified by sex. Additional adjustment for follow-up-time and BMI criteria used (above 50 kg/m$^2$ or above 35 kg/m$^2$ with obesity-related complications) was performed, but excluded from the final presentation as it did not substantially change the main results.

P-values below 0.05 were considered as statistically significant. Data was analyzed with SPSS version 25 for Windows (Armonk, NY).

## Ethics

The study was conducted in accordance with the Declaration of Helsinki and was approved by the Scientific Ethics Committee of the Capital Region, Denmark, protocol number HD2009-78. Informed consent was obtained from all individual participants included in the study.

## Results

A total of 576 patients (74% females) with a mean age of 45.4 ± SD 10.3 years and a mean BMI of 44.3 ± 5.2 kg/m$^2$ were included. The mean change in BMI from inclusion to the nadir 1–3 years after bariatric surgery was 14.4 kg/m$^2$, corresponding to an average weight loss of 42 kg (Table 1).

There was a direct association between baseline BMI and weight loss in kg (β = 1.22 kg/[kg/m$^2$], 95% CI 1.03 to 1.41, p<0.001). In the subset of 272 patients with measured baseline WHR, the mean WHR was 0.95 ± 0.11. We observed an inverse association between baseline WHR$_{BMI}$ and weight loss in kg (β = -11.8 kg/unit, 95% CI -23.3 to -0.3, p = 0.047) driven by a particularly strong effect in women (S1 Table).

The genetic determinants examined in the study are given in Table 2.

Mean GRS-BMI$_{941}$ was 913 ± 17 risk alleles, and mean GRS-WHR$_{344}$ was 342 ± 10 risk alleles (Table 1). The GRS-BMI$_{941}$ was not significantly associated with BMI or WHR$_{BMI}$ at baseline. The GRS-WHR$_{344}$ was inversely associated with baseline BMI (β = -5.5 kg/m$^2$/100 risk alleles, 95% CI -9.7 to -1.2, p = 0.01) and positively associated with WHR$_{BMI}$ (β = 0.15 units/100 risk alleles, 95% CI 0.01 to 0.29, p = 0.04). *VEGFA*-rs1358980-T was not associated with BMI or WHR$_{BMI}$ at baseline (Table 3).

The GRS-BMI$_{941}$ was not significantly associated with the weight loss response to bariatric surgery in the cohort as a whole (β = -1.3 kg/100 risk alleles, 95% CI -7.5 to 4.9, p = 0.68). A high GRS-WHR$_{344}$ was associated with an attenuated weight loss before adjustment for baseline BMI (β = -14.6 kg/100 risk alleles, 95% CI -25.4 to -3.8, p = 0.008), but not after adjustment for baseline BMI (β = -7.9 kg/100 risk alleles, 95% CI -17.5 to 1.6, p = 0.11). The

**Table 1. Demographics, clinical characteristics, and genetic determinants in the cohort, stratified by sex.**

| | Whole group | Female | Male |
|---|---|---|---|
| Included patients, n | 576 | 428 (74%) | 148 (26%) |
| Center, n | | | |
| Hvidovre | 351 (61%) | 250 (71%) | 101 (29%) |
| Hamlet | 225 (39%) | 178 (79%) | 47 (21%) |
| Age (years) | 45.4 (10.3) | 45.0 (10.3) | 46.4 (10.3) |
| Height (cm) | 170.6 (9.4) | 166.8 (6.6) | 181.5 (7.7) |
| Waist circumference (cm) | 127.9 (14.4) | 123.0 (12.3) | 140.2 (11.5) |
| Hip circumference (cm) | 134.9 (13.1) | 136.6 (13.0) | 130.6 (12.5) |
| Waist-hip-ratio | 0.95 (0.11) | 0.90 (0.08) | 1.08 (0.08) |
| Initial weight (kg) | 129.5 (21.2) | 123.2 (17.3) | 147.8 (20.6) |
| Nadir weight (kg) | 87.5 (18.3) | 81.8 (14.8) | 103.9 (17.8) |
| Change in weight (kg) | 42.0 (13.3) | 41.3 (12.6) | 43.9 (15.1) |
| Weight loss (%) | 32.4 (8.5) | 33.4 (8.3) | 29.5 (8.5) |
| Initial BMI (kg/m$^2$) | 44.3 (5.2) | 44.2 (5.1) | 44.9 (5.5) |
| Nadir BMI (kg/m$^2$) | 29.9 (4.9) | 29.4 (4.8) | 31.5 (4.9) |
| BMI loss (kg/m$^2$) | 14.4 (4.4) | 14.8 (4.3) | 13.3 (4.5) |
| GRS-BMI$_{941}$ * | 913 (17) | 912 (17) | 914 (17) |
| GRS-WHR$_{344}$ * | 342 (10) | 342 (10) | 340 (10) |
| VEGFA-rs1358980 | | | |
| 0 risk alleles | 167 (29%) | 123 (29%) | 44 (30%) |
| 1 risk allele | 283 (49%) | 216 (50%) | 67 (45%) |
| 2 risk alleles | 126 (22%) | 90 (21%) | 37 (25%) |

Data from 576 patients are included in all rows, except waist circumference, hip circumference and waist-hip-ratio where 272 patients are included. BMI; Body mass index. GRS; Genetic risk score. WHR; Waist-hip-ratio.
*weighted sum of risk alleles.
Presented as means with standard deviations within parentheses; alternatively as numbers with percentages within parentheses.

VEGFA-rs1358980-T risk allele was not significantly associated with weight loss ($\beta$ = -1.4 kg/risk allele, 95% CI -2.9 to 0.1, p = 0.07) (Table 4).

## Discussion

Neither was the GRS-BMI$_{941}$ associated with the weight loss response after bariatric surgery in the total cohort nor was the GRS-WHR$_{344}$ associated with an attenuated weight loss response to bariatric surgery after adjustment for baseline BMI. Similarly, we did not find a significant association between the *VEGFA*-rs1358980-T risk allele and the weight loss response with or without adjustment for baseline BMI in the total cohort.

The GRS-BMI$_{941}$ was not associated with baseline BMI in our cohort, neither in the total cohort nor in men or women separately. This could be due to a truncation effect, as patients in bariatric surgery cohorts are included based on whether BMI is above a specific threshold. Additionally, as the SNPs used in the GRS-BMI$_{941}$ are based on the BMI distribution in the general population, the association between this GRS and BMI might not be present among patients in in the extreme upper end of the BMI distribution. In line with previous studies using GRSs based on fewer SNPs [11, 14] we also failed to show any links between the GRS-BMI$_{941}$ and weight loss response following bariatric surgery.

**Table 2. Genetic determinants examined in the manuscript.**

| Genetic exposure | Description | Reference | Number of SNPs |
|---|---|---|---|
| GRS-BMI$_{941}$ | Genetic risk score for BMI based on meta-analysis | *Yengo et al.* 2018 [15] | 941 |
| GRS-WHR$_{344}$ | Genetic risk score for abdominal obesity based on meta-analysis of SNPs associated with WHR adjusted for BMI | *Pulit et al.* 2019 [17] | 344 |
| VEGFA-rs1358980 | SNP of special interest in the analysis of weight changes | *Shungin et al. 2015* [21] *Svendstrup et al. 2018* [16] | 1 |

BMI; Body mass index. WHR; Waist-hip-ratio. SNP; Single nucleotide polymorphism.

The GRS-WHR$_{344}$ was as expected positively associated with WHR$_{BMI}$ at baseline in the total cohort as well as when both sexes were analyzed separately [17]. However, the GRS-WHR$_{344}$ was further (somewhat surprising) inversely associated with BMI at baseline. A possible explanation could be the genetic pathways linking some genetic markers of abdominal obesity with a relatively lower BMI as suggested by Lu et al [22]. The association pattern might also be related to the so-called 'adipose expandability hypothesis' suggesting abdominal obesity to be a result of a reduced ability to store fat subcutaneously, leading to visceral and ectopic fat storing further eliciting insulin resistance and other dys-metabolic traits [20]. In other words, a reduced capability of adipose tissue expansion leads to a more dys-metabolic abdominal obesity at a lower BMI threshold. Patients with abdominal obesity and metabolic disturbances (e.g. type 2 diabetes) will be accepted for bariatric surgery at a lower BMI due to the applied selection criteria for surgery in Denmark [23]. This practice might explain the inverse association between GRS-WHR$_{344}$ and BMI in the current cohort of patients accepted for bariatric surgery.

The association between GRS-WHR$_{344}$ and attenuated weight loss after bariatric surgery is supported by previous comparable findings in data of non-surgical weight loss from the NUGENOB study [16]. However, in contrast to the findings in the NUGENOB study, the significance of the association in the present study was lost after adjustment for baseline BMI.

**Table 3. Associations between genetic determinants and anthropometric traits at baseline.**

| Genetic exposure | Association with BMI at baseline[1] | | | Association with WHR$_{BMI}$ at baseline[2] | | |
|---|---|---|---|---|---|---|
| | β | 95% CI | p | β | 95% CI | p |
| | Whole group | | | | | |
| GRS-BMI$_{941}$ | 0.7 | -1.7 to 3.2 | 0.56 | 0.00 | -0.07 to 0.08 | 0.97 |
| GRS-WHR$_{344}$ | -5.5 | -9.7 to -1.2 | 0.01 | 0.15 | 0.01 to 0.29 | 0.04 |
| VEGFA-rs1358980 | -0.5 | -1.1 to 0.1 | 0.11 | 0.00 | -0.02 to 0.02 | 0.96 |
| | Women alone | | | | | |
| GRS-BMI$_{941}$ | 0.5 | -2.3 to 3.3 | 0.73 | -0.04 | -0.10 to 0.03 | 0.26 |
| GRS-WHR$_{344}$ | -1.5 | -6.3 to 3.3 | 0.54 | 0.14 | 0.02 to 0.26 | 0.02 |
| VEGFA-rs1358980 | -0.3 | -1.0 to 0.4 | 0.44 | 0.01 | -0.01 to 0.03 | 0.31 |
| | Men alone | | | | | |
| GRS-BMI$_{941}$ | 1.1 | -4.2 to 6.4 | 0.68 | -0.05 | -0.15 to 0.04 | 0.27 |
| GRS-WHR$_{344}$ | -16.3 | -25.1 to -7.4 | <0.001 | 0.24 | 0.06 to 0.41 | 0.01 |
| VEGFA-rs1358980 | -1.1 | -2.3 to 0.2 | 0.09 | -0.01 | -0.04 to 0.01 | 0.31 |

BMI; Body mass index. CI; Confidence interval. GRS; Genetic risk score. WHR; Waist-hip-ratio. WHR$_{BMI}$; Waist-hip-ratio adjusted for BMI.

Linear regression with respectively BMI (kg/m$^2$; n = 576) and WHR adjusted for BMI (unit-free; n = 272) as dependent variable. The presented β values are effects per 100 risk alleles (GRS-BMI$_{941}$ and GRS-WHR$_{344}$) or effects per risk allele (VEGFA-rs1358980).

**Table 4. Genetic determinants of change in weight after bariatric surgery.**

| Genetic exposure | Model 1 | | | Model 2 | | |
|---|---|---|---|---|---|---|
| | β | 95% CI | p | β | 95% CI | p |
| | Whole group | | | | | |
| GRS-BMI$_{941}$ | -1.3 | -7.5 to 4.9 | 0.68 | -2.0 | -7.5 to 3.3 | 0.45 |
| GRS-WHR$_{344}$ | -14.6 | -25.4 to -3.8 | 0.008 | -7.9 | -17.5 to 1.6 | 0.11 |
| VEGFA- rs1358980 | -1.4 | -2.9 to 0.1 | 0.07 | -0.8 | -2.2 to 0.6 | 0.25 |
| | Women alone | | | | | |
| GRS-BMI$_{941}$ | 2.4 | -4.4 to 9.1 | 0.49 | 1.8 | -4.1 to 7.7 | 0.54 |
| GRS-WHR$_{344}$ | -8.7 | -20.6 to 3.2 | 0.15 | -6.8 | -17.2 to 3.6 | 0.20 |
| VEGFA- rs1358980 | -0.5 | -2.2 to 1.2 | 0.56 | -0.18 | -1.7 to 1.3 | 0.81 |
| | Men alone | | | | | |
| GRS-BMI$_{941}$ | -11.4 | -25.9 to 3.1 | 0.12 | -13.7 | -26.4 to -1.0 | 0.04 |
| GRS-WHR$_{344}$ | -30.8 | -55.2 to -6.3 | 0.01 | -10.1 | -33.1 to 12.9 | 0.39 |
| VEGFA- rs1358980 | -3.5 | -6.9 to -0.2 | 0.04 | -2.34 | -5.33 to 0.65 | 0.13 |

BMI; Body mass index. CI; Confidence interval. GRS; Genetic risk score. WHR; Waist-hip-ratio.

Linear regression with weight change as dependent variable ($\Delta$kg; n = 576). The presented β values are effects per 100 risk alleles (GRS-BMI$_{941}$ and GRS-WHR$_{344)}$ or per risk allele (VEGFA-rs1358980).

Model 1: adjustments for sex, age and inclusion center. Model 2: adjustments for sex, age, inclusion center and baseline BMI.

Reported studies of patients undergoing bariatric surgery have also pointed to relationships between weight loss and genetic markers associated with abdominal obesity. Notably, these studies used excess BMI loss as the weight loss phenotype, a variable that is strongly related to the baseline BMI [6, 11]. Our data indicate that the association between GRS-WHR$_{344}$ and weight loss response to bariatric surgery might be explained by the association between the GRS-WHR$_{344}$ and baseline anthropometrics, especially BMI [17, 21]. Baseline BMI is known to be associated with the response to weight loss interventions. The choice of weight loss phenotype is therefore of great importance in this type of study. For mathematical reasons, the excess body mass index and excess weight loss variables are inversely associated with baseline BMI in bariatric surgery cohorts, while BMI change or weight change are positively associated with baseline BMI [24].

The polymorphisms related to body fat distribution have been suggested to mechanistically act in adipose tissue in three main pathways: Adipogenesis, angiogenesis and non-specified transcriptional regulation [21]. Adipogenesis and angiogenesis are important for adipose tissue function and expansion and impairment of these mechanisms lead to metabolic disturbances through induction of hypoxia, inflammation and fibrosis in the tissue [25–29]. Human studies have further shown that fibrosis and reduced angiogenesis in adipose tissue may attenuate weight loss response after gastric bypass [30–32]. Therefore, it may be hypothesized that various genetic factors determinant for abdominal obesity and for weight loss responsiveness following surgical interventions may operate via common pathways of adipogenesis and angiogenesis.

We investigated the *VEGFA*-rs1358980-T risk allele as a potential candidate SNP for angiogenesis. Previously, we reported that the variant attenuated weight loss in women in a diet-induced weight loss program in the NUGENOB study [16]. The polymorphism is located at a genome position upstream to *VEGFA*, an angiogenesis gene encoding the VEGF-A protein. The protein induces angiogenesis and elevated plasma and adipose tissue concentrations of VEGF-A in obesity are known to decline following weight loss [32, 33]. In the present study the *VEGFA*-rs1358980-T risk allele was not significantly associated with BMI or WHR$_{BMI}$ at

baseline. The inverse effect direction of *VEGFA*-rs1358980-T on the weight loss response was consistent with the findings from the NUGENOB study [16]. However, we cannot exclude that lack of significance between *VEGFA*-rs1358980-T and baseline WHR$_{BMI}$ as well as with weight loss in the present study might be an issue of statistical power.

The application of GRSs to clinical trials is of no direct clinical relevance at the present due to very small effect sizes of GRS. Still, such studies are crucial for understanding potential genetic linkages between traits and thereby for fostering novel hypothesis for biological pathways behind shared phenotypes to be tested in subsequent mechanistic experiments. Angiogenesis may be one of the mechanisms that govern the individual variation in response to weight loss treatment by possibly affecting adipose tissue flexibility. The weight change after bariatric surgery might be associated with genetic factors associated with weight change rather than SNPs associated with absolute values of BMI and for instance the putative role of *VEGFA* should be explored further; one approach could be a meta-analysis of weight loss following interventions related to diet, physical activity, drugs or bariatric surgery [34].

Finally, it should be borne in mind that a multitude of factors influence weight loss after bariatric surgery, like procedure of surgery, postprandial gastrointestinal hormone responses, diabetes status, preoperative eating disturbances, postoperative eating pattern and adaptive changes of the intestinal microbiota [35–37]. The many factors makes it challenging to disentangle the different effects including the potential roles of the GRSs.

## Strengths and limitations

The main strength of this study is that the evaluated GRSs are established risk scores based on large population studies, rather than GRSs that have been constructed and tested in the same cohort. Other strengths include standardized surgery, genotyping of high quality and appropriate methods for imputation of genotypes. The cohort is larger than many of the comparable studies examining GRSs for weight loss following bariatric surgery, but smaller than genetic studies in general. An additional strength is that the patients are investigated 1–3 years after surgery, around the time point with highest weight loss.

WHR was only measured in half of the patients at baseline, which limits the exploration of the data due to reduced statistical power. Three different genetic markers were evaluated without correction for multiple testing; if applied, this would not in any substantial way affect the main interpretations given the general non-significance nature of the results. The measurement of hip and waist circumference might be slightly imprecise among the patients with highest BMI. While the loss to follow up is a third weakness that could lead to bias, this is not expected to mainly be caused by the genetics of the patients and should therefore not affect the main conclusions of the study.

## Conclusions

With an updated genome-wide GRS for BMI and with a genome-wide GRS for abdominal obesity we were unable to demonstrate any significant relationships between genetics and weight loss following bariatric surgery in morbidly obese patients. Similarly, the rs1358980-T risk allele in the *VEGFA* locus was not linked with an attenuated weight loss. Future developments of clinically relevant GRS for predicting weight loss after various interventions including bariatric surgery are dependent on progress in the identification of specific genome regions related to weight loss *per se* to complement current knowledge of the genetics behind various adiposity measures.

## Supporting information

**S1 Table. Associations between baseline anthropometrics and weight loss after bariatric surgery.**
(DOCX)

## Author Contributions

**Conceptualization:** Mathilde Svendstrup, Theresia M. Schnurr, Dorte Lindqvist Hansen, Dorte Worm, Niels Grarup, Henrik Vestergaard, Oluf Pedersen, Lars Ängquist, Thorkild I. A. Sørensen, Sten Madsbad, Torben Hansen.

**Data curation:** Martin Aasbrenn, Mathilde Svendstrup, Theresia M. Schnurr, Dorte Lindqvist Hansen, Dorte Worm, Marie Balslev-Harder, Niels Grarup, Kristoffer Sølvsten Burgdorf, Henrik Vestergaard.

**Formal analysis:** Martin Aasbrenn, Mathilde Svendstrup, Theresia M. Schnurr, Lars Ängquist, Sten Madsbad, Torben Hansen.

**Funding acquisition:** Sten Madsbad, Torben Hansen.

**Investigation:** Martin Aasbrenn, Mathilde Svendstrup, Theresia M. Schnurr, Dorte Lindqvist Hansen, Dorte Worm, Marie Balslev-Harder, Oluf Pedersen, Mogens Fenger, Thorkild I. A. Sørensen, Sten Madsbad, Torben Hansen.

**Methodology:** Mathilde Svendstrup, Theresia M. Schnurr, Niels Grarup, Kristoffer Sølvsten Burgdorf, Henrik Vestergaard, Lars Ängquist, Mogens Fenger, Thorkild I. A. Sørensen, Sten Madsbad, Torben Hansen.

**Project administration:** Martin Aasbrenn, Mathilde Svendstrup, Dorte Lindqvist Hansen, Dorte Worm, Marie Balslev-Harder, Niels Grarup, Kristoffer Sølvsten Burgdorf, Henrik Vestergaard, Sten Madsbad, Torben Hansen.

**Supervision:** Mathilde Svendstrup, Theresia M. Schnurr, Henrik Vestergaard, Oluf Pedersen, Lars Ängquist, Thorkild I. A. Sørensen, Sten Madsbad, Torben Hansen.

**Validation:** Mathilde Svendstrup, Torben Hansen.

**Writing – original draft:** Martin Aasbrenn, Mathilde Svendstrup, Theresia M. Schnurr, Dorte Lindqvist Hansen, Dorte Worm, Marie Balslev-Harder, Niels Grarup, Kristoffer Sølvsten Burgdorf, Henrik Vestergaard, Oluf Pedersen, Lars Ängquist, Mogens Fenger, Thorkild I. A. Sørensen, Sten Madsbad, Torben Hansen.

**Writing – review & editing:** Martin Aasbrenn, Mathilde Svendstrup, Theresia M. Schnurr, Dorte Lindqvist Hansen, Dorte Worm, Marie Balslev-Harder, Niels Grarup, Kristoffer Sølvsten Burgdorf, Henrik Vestergaard, Oluf Pedersen, Lars Ängquist, Mogens Fenger, Thorkild I. A. Sørensen, Sten Madsbad, Torben Hansen.

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
