## [Decision Letter · Decision Letter 0]

15 Jan 2021

PONE-D-20-36337

Genetic markers of abdominal obesity and weight loss after gastric bypass surgery

PLOS ONE

Dear Dr. Aasbrenn,

Thank you for submitting your manuscript to PLOS ONE. After careful consideration, we feel that it has merit but does not fully meet PLOS ONE’s publication criteria as it currently stands. Therefore, we invite you to submit a revised version of the manuscript that addresses the points raised during the review process.

We look forward to receiving your revised manuscript.

Kind regards,

Sabine Rohrmann

Academic Editor

PLOS ONE

Journal Requirements:

"This study was funded by a grant from The Ministry of Higher Education and Science The UNIK

317 Initiative: Food, Fitness & Pharma for Health and Disease. The project was also supported with funds

318 from the Challenge Programme MicrobLiver (Grant No. NNF15OC0016692). The Novo Nordisk

319 Foundation Center for Basic Metabolic Research is an independent research centre at the University of

320 Copenhagen and is partly funded by an unrestricted donation from the Novo Nordisk Foundation."

"This study was funded by a grant from The Ministry of Higher Education and Science  The UNIK Initiative: Food, Fitness & Pharma for Health and Disease (TH, https://ufm.dk/). The project was also supported with funds from the Challenge Programme MicrobLiver (Grant No. NNF15OC0016692) (TH, https://challenge.novonordiskfoundation.com/). The funders had no role in study design, data collection and analysis, decision to publish, or preparation of the manuscript."

Reviewers' comments:

Reviewer's Responses to Questions

**Comments to the Author**

1. Is the manuscript technically sound, and do the data support the conclusions?

Reviewer #1: Partly

2. Has the statistical analysis been performed appropriately and rigorously? 

Reviewer #1: Yes

3. Have the authors made all data underlying the findings in their manuscript fully available?

Reviewer #1: No

4. Is the manuscript presented in an intelligible fashion and written in standard English?

Reviewer #1: Yes

5. Review Comments to the Author

Reviewer #1: This study appears to address two main questions. First, is a genetic risk score (GRS) derived from hundreds of genes related to obesity associated with the amount of weight lost after gastric bypass surgery. Second, is a specific SNP in the VGFA locus that has been related to weight loss in non-surgical weight loss intervention studies associated with BMI or waist-hip ratio (WHR) adjusted for BMI in their cohort. Both are important questions to address since replication in genetic studies across populations is so important.

The sample size of gastric bypass patients is large with 576 subjects but may not be large enough to detect individual SNP associations with small effects, such as VGFA, but is likely large enough for power to detect associations with a GRS. The cohort is described well, but there was a large number (N=487) of the initial 1511 subjects excluded for reasons related to QC of the genotype data. The listed reasons for exclusion would normally not exclude that many subjects. Perhaps the N for each reason could be added to give a clearer picture of the quality of the genotyping. The remaining 448 subjects didn’t return for a follow-up exam. Were their baseline BMIs or ages different than those that participated? Subjects with more successful weight loss might be more prone to come back for a second exam and might be healthier, both of which could possibly bias the outcome.

It is not clear to me what the term ‘clinical control’ refers to in the methods section (the paragraph under the Anthropometrics heading. Does this mean clinical sites or clinical visits?

Reading the abstract, it appears that the authors are trying to put a positive message out that there are some statistically significant findings that may be important. However, the abstract makes the results seem weak or borderline. This is my major concern with how the paper is currently written. Considering that there were three genetic variables (BMI GRS, WHR GRS, VGFA SNP) and two sexes tested, there is a minimum of 6 multiple comparisons that require adjustment of the p-values. If one makes the appropriate corrections to the results, perhaps only one finding is barely significant. The VGFA ‘borderline’ finding of p=0.07 with weight loss goes away, failing to replicate other studies. While in men it was associated at p=0.04, this was before adjustment for baseline BMI even without multiple testing correction.

Given the importance of the two questions being addressed by this study, my interpretation of the results is that a GRS derived from large meta-analyses for either BMI or WHR does not predict weight loss after gastric bypass surgery, especially when one corrects for baseline BMI. Baseline BMI is known to be associated with subsequent surgical weight loss and implicating genes involved with weight loss would seem to require them to be independent of baseline BMI (which in this study was not in and of itself related to the GRS for probable reasons listed by the authors). Therefore, unless the authors believe there really is a significant finding that they can better defend, my suggestion would be to re-write the paper as a negative study for the usefulness of these GRSs or VGFA to predict weight loss. This would still be an important contribution, since GRS seems to be in vogue to represent the polygenic nature of obesity.

A minor suggestion would be to actually test for the sex difference in results related to line 187. Then a p-value could replace ‘might be stronger among the men than the women.’ If not significant, this phrase should be removed.

The hypothesis raised about more abdominal obesity and metabolic abnormalities with lower BMI seems to require some non-linear function of BMI (lines 234-238), as one wouldn’t expect a continually lower BMI, say down to 25, to further increase metabolic abnormalities. So this hypothesis requires only a middle range of BMI to be unhealthy with high ranges or low ranges to be less unhealthy. But high BMI ranges should also have reduced capability of adipose tissue expansion. Or did I miss the argument here? In any case, the strength of the results (weak and inconsistent) seems to make it advisable not to promote a new hypothesis to explain these weak results as in lines 259-261.

The statement in lines 276-277 that weight change genes may differ from BMI genes may be true. However, an alternative hypothesis might be that genes the increase BMI in a population would likely lead to resistance to weight loss. So maybe a better explanation would be that surgical effects of weight loss are so strong as to over-ride the smaller genetic effects resisting weight loss or even a different set of weight-change SNPs.

Line 293. In addition to hip measurement variability, doesn’t the same concern apply to waist circumference? There will be variability in waist circumference too by measuring wherever the largest waist occurred, as was done, rather than at some standard point like just above the iliac crest.

6. PLOS authors have the option to publish the peer review history of their article (what does this mean?). If published, this will include your full peer review and any attached files.

Reviewer #1: No

---

## [Author Response · Author response to Decision Letter 0]

22 Mar 2021

PONE-D-20-36337

Genetic markers of abdominal obesity and weight loss after gastric bypass surgery

PLOS ONE

Dear Editor,

This response explains point-by-point how we have changed the manuscript after the very useful comments from the editor and the reviewer.

Comments from the Editor 

1. We have adapted the manuscript to PLOS ONE´s style requirements.

2. The funding information has been removed from the revised manuscript. In the online submission form, we prefer this amended statement: 

“This study was funded by a grant from The Ministry of Higher Education and Science The UNIK Initiative: Food, Fitness & Pharma for Health and Disease (TH, http://ufm.dk). The project was also supported with funds from the Challenge Programme MicrobLiver (Grant No. NNF15OC0016692) (TH, http://challenge.novonordiskfoundation.com). The Novo Nordisk Foundation Center for Basic Metabolic Research is an independent research centre at the University of Copenhagen and is partly funded by an unrestricted donation from the Novo Nordisk Foundation.”

3. The data set contains potentially identifying and sensitive genetic information on a vulnerable population, and the permission from the Scientific Ethics Committee of the Capital Region, Denmark from 2009 did therefore not include acceptance of sharing of data. Anonymized patient level data can only legally be accessed upon acceptance from The Danish Data Protection Agency (www.datatilsynet.dk; dt@datatilsynet.dk), the goverment body that regulates the use of Danish research data. Interested parties can contact the corresponding author (MA) for further information.

4. We have amended the list of authors on the manuscript and added affiliations.

Comments from the Reviewer

Questions from the reviewer have been numbered. 

Each question is followed by our answer, with references to the lines where changes have been made in the manuscript. 

1.The sample size of gastric bypass patients is large with 576 subjects but may not be large enough to detect individual SNP associations with small effects, such as VGFA, but is likely large enough for power to detect associations with a GRS. The cohort is described well, but there was a large number (N=487) of the initial 1511 subjects excluded for reasons related to QC of the genotype data. The listed reasons for exclusion would normally not exclude that many subjects. Perhaps the N for each reason could be added to give a clearer picture of the quality of the genotyping. 

1. Before running quality control (QC) analyses, a total of 1511 samples from 1414 individuals and 538,448 SNPs were available. 97 of these samples were from the same individuals; as samples from some individuals were analyzed more than once to utilize all the genotyping capacity. The previous number of 1511 participants on line 108 is therefore in hindsight incorrect, this has been to corrected to 1414.

After QC, a total of 1024 individuals and 241,667 SNPs were eligible for further investigation. Prior to genotype imputation, samples were excluded using following criteria:

i) a genotype call-rate below 95% (removed n=177 individuals)

ii) extreme positive or negative inbreeding coefficients (removed n=99 individuals)

iii) individuals of divergent ancestry using Principal Component Analysis (PCA) (removed n=72 individuals)

iv) first degree relative relations found by Identical By Descent analysis where only the relative with the highest call-rate for each pedigree pair was retained (removed n=34 individuals)

v) sex discordant information between genotype and phenotype data (removed n=8 individuals)

The N for each reason ihas been added on line 112-114.

2. The remaining 448 subjects didn’t return for a follow-up exam. Were their baseline BMIs or ages different than those that participated? Subjects with more successful weight loss might be more prone to come back for a second exam and might be healthier, both of which could possibly bias the outcome.

2. The subjects who did not return had a mean baseline BMI of 46.1 kg/m2 (SD 7.1), in the same range as the included patients (mean 45.4 kg/m2 (SD 10.3). The subjects who did not return were slightly younger (mean 43.9 years, SD 9.3) than the subjects who were included (mean 45.4 years, SD 10.3). 

 It had indeed been interesting to know the weight loss of the subjects that did not come back, but unfortunately this was not available in our data set.

3.It is not clear to me what the term ‘clinical control’ refers to in the methods section (the paragraph under the Anthropometrics heading. Does this mean clinical sites or clinical visits?

3.. It should have been “clinical visit”. On line 133, we have therefore changed the text to “The lowest body weight at a post-surgical clinical visit”

4. Reading the abstract, it appears that the authors are trying to put a positive message out that there are some statistically significant findings that may be important. However, the abstract makes the results seem weak or borderline. This is my major concern with how the paper is currently written. Considering that there were three genetic variables (BMI GRS, WHR GRS, VGFA SNP) and two sexes tested, there is a minimum of 6 multiple comparisons that require adjustment of the p-values. If one makes the appropriate corrections to the results, perhaps only one finding is barely significant. The VGFA ‘borderline’ finding of p=0.07 with weight loss goes away, failing to replicate other studies. While in men it was associated at p=0.04, this was before adjustment for baseline BMI even without multiple testing correction.

4. We do agree with this comment, and have rewritten the abstract (30-56) and the conclusion (lines 309-315) in addition to several small changes throughout the results and discussion paragraphs. The lack of adjustment for multiple testing is added to the limitations (lines 303-304)

5. Therefore, unless the authors believe there really is a significant finding that they can better defend, my suggestion would be to re-write the paper as a negative study for the usefulness of these GRSs or VGFA to predict weight loss. This would still be an important contribution, since GRS seems to be in vogue to represent the polygenic nature of obesity.

5. We do find this interpretation reasonable and have rewritten the abstract, discussion and conclusions as a negative study (lines 30-56 and 220-315). We are happy to hear that the reviewer agrees with us the research questions are important.

6. A minor suggestion would be to actually test for the sex difference in results related to line 187. Then a p-value could replace ‘might be stronger among the men than the women.’ If not significant, this phrase should be removed.

6. As the associations at baseline are not the main scope of the article, we do prefer to remove this phrase (line193).

7. The hypothesis raised about more abdominal obesity and metabolic abnormalities with lower BMI seems to require some non-linear function of BMI (lines 234-238), as one wouldn’t expect a continually lower BMI, say down to 25, to further increase metabolic abnormalities. So this hypothesis requires only a middle range of BMI to be unhealthy with high ranges or low ranges to be less unhealthy. But high BMI ranges should also have reduced capability of adipose tissue expansion. Or did I miss the argument here? 

7. The idea is that patients in the ‘lower BMI end’ (BMI < 40) are more metabolically burdened by their overweight, otherwise they wouldn’t have been selected for RYGB at all. So we’re perhaps dealing with two different phenotypes: 1) the very obese but relatively healthy (considered their extreme obesity) patients and 2) the less obese but relatively unhealthy patients. This fits very well in the adipose expandability hypothesis. We have tried to explain this a bit further on line 242-245, adding the point about the selection criteria.

8. In any case, the strength of the results (weak and inconsistent) seems to make it advisable not to promote a new hypothesis to explain these weak results as in lines 259-261.

8. The statement (now on lines 266-268) has been rephrased.

9. The statement in lines 276-277 that weight change genes may differ from BMI genes may be true. However, an alternative hypothesis might be that genes that increase BMI in a population would likely lead to resistance to weight loss. So maybe a better explanation would be that surgical effects of weight loss are so strong as to over-ride the smaller genetic effects resisting weight loss or even a different set of weight-change SNPs.

9. It is possible that genes that increase BMI in a population could lead to resistance to weight loss. When we do not find these relations in our bariatric surgery cohort, it might be due to a selection bias: It is possible that subjects with low genetic risk who nevertheless become morbidly obese might have other causes for obesity (e.g. social) that also attenuate the response to bariatric surgery. Unfortunately, we do not have information about this in our data set. We have tried to introduce these perspectives on lines 288-292.

10. Line 293. In addition to hip measurement variability, doesn’t the same concern apply to waist circumference? There will be variability in waist circumference too by measuring wherever the largest waist occurred, as was done, rather than at some standard point like just above the iliac crest.

10. The risk of variability in waist circumference measurements has been added (line 305).

Yours sincerely, on behalf of all authors,

Martin Aasbrenn

---

## [Decision Letter · Decision Letter 1]

11 Apr 2021

PONE-D-20-36337R1

Genetic markers of abdominal obesity and weight loss after gastric bypass surgery

PLOS ONE

Dear Dr. Aasbrenn,

Thank you for submitting your manuscript to PLOS ONE. After careful consideration, we feel that it has merit but does not fully meet PLOS ONE’s publication criteria as it currently stands. Therefore, we invite you to submit a revised version of the manuscript that addresses the points raised during the review process.

or perceived impact.

We look forward to receiving your revised manuscript.

Kind regards,

Sabine Rohrmann

Academic Editor

PLOS ONE

Journal Requirements:

Reviewers' comments:

Reviewer's Responses to Questions

**Comments to the Author**

1. If the authors have adequately addressed your comments raised in a previous round of review and you feel that this manuscript is now acceptable for publication, you may indicate that here to bypass the “Comments to the Author” section, enter your conflict of interest statement in the “Confidential to Editor” section, and submit your "Accept" recommendation.

Reviewer #1: All comments have been addressed

2. Is the manuscript technically sound, and do the data support the conclusions?

Reviewer #1: Yes

3. Has the statistical analysis been performed appropriately and rigorously? 

Reviewer #1: Yes

4. Have the authors made all data underlying the findings in their manuscript fully available?

Reviewer #1: Yes

5. Is the manuscript presented in an intelligible fashion and written in standard English?

Reviewer #1: Yes

6. Review Comments to the Author

Reviewer #1: The authors have addressed my comments on the manuscript. I found two grammatical changes to fix and have one additional suggestion. Line 255 (in the non-tracked version) reads "this type of studies." It should be changed to "this type of study." Line 115 should be corrected to something like "did not have the required follow-up visits." The authors added a statement about not adjusting for multiple comparisons on lines 302 and 303. Since there is really no result that is "hypothesis generating," I would prefer that the statement be used in positive support of their findings. I would state that multiple comparison adjustments were not applied to the three genetic risk factor p-values, but if applied they would make the results even less significant than reported, further supporting the conclusions of the study.

7. PLOS authors have the option to publish the peer review history of their article (what does this mean?). If published, this will include your full peer review and any attached files.

Reviewer #1: No

---

## [Author Response · Author response to Decision Letter 1]

4 May 2021

PONE-D-20-36337

Genetic markers of abdominal obesity and weight loss after gastric bypass surgery

PLOS ONE

Dear Editor,

This response explains point-by-point how we have changed the manuscript after the additional useful comments from the reviewer. We have also corrected some inconsistencies related to table footnotes, number of decimals in table 3 and 4, punctuation, word choice and font size.

Changes in the text are highlighted with blue font color in the file labeled “Revised Manuscript with Track Changes”. 

Comments from the Reviewer

1. Line 255 (in the non-tracked version) reads "this type of studies." It should be changed to "this type of study." 

We have changed this (line 252).

2. Line 115 should be corrected to something like "did not have the required follow-up visits." 

We have corrected this sentence (line 115).

3. The authors added a statement about not adjusting for multiple comparisons on lines 302 and 303. Since there is really no result that is "hypothesis generating," I would prefer that the statement be used in positive support of their findings. I would state that multiple comparison adjustments were not applied to the three genetic risk factor p-values, but if applied they would make the results even less significant than reported, further supporting the conclusions of the study.

This is a very good point, and we have removed the sentence about potential hypothesis generating findings. We agree that the main interpretation of the study (e.g. negative) would be the same if correction for multiple testing were applied (lines 300-302).

We look forward to hear from you again.

Yours sincerely, on behalf of all authors,

Martin Aasbrenn

---

## [Editor Report · Decision Letter 2]

18 May 2021

Genetic markers of abdominal obesity and weight loss after gastric bypass surgery

PONE-D-20-36337R2

Dear Dr. Aasbrenn,

We’re pleased to inform you that your manuscript has been judged scientifically suitable for publication and will be formally accepted for publication once it meets all outstanding technical requirements.

Kind regards,

Sabine Rohrmann

Academic Editor

PLOS ONE
---

## [Editor Report · Acceptance letter]

20 May 2021

PONE-D-20-36337R2 

Genetic markers of abdominal obesity and weight loss after gastric bypass surgery 

Dear Dr. Aasbrenn:

I'm pleased to inform you that your manuscript has been deemed suitable for publication in PLOS ONE. Congratulations! Your manuscript is now with our production department. 

Kind regards, 

on behalf of

Dr. Sabine Rohrmann 

Academic Editor

PLOS ONE